# Towards Zebrafish Models of CNS Channelopathies

**DOI:** 10.3390/ijms232213979

**Published:** 2022-11-12

**Authors:** Tatiana O. Kolesnikova, Konstantin A. Demin, Fabiano V. Costa, Konstantin N. Zabegalov, Murilo S. de Abreu, Elena V. Gerasimova, Allan V. Kalueff

**Affiliations:** 1Neurobiology Program, Sirius University of Science and Technology, 354349 Sochi, Russia; 2Institute of Translational Biomedicine, St. Petersburg State University, 199034 St. Petersburg, Russia; 3Institute of Experimental Medicine, Almazov National Medical Research Centre, Ministry of Healthcare of Russian Federation, 197341 St. Petersburg, Russia; 4Moscow Institute of Physics and Technology, 141701 Moscow, Russia; 5Laboratory of Preclinical Bioscreening, Granov Russian Research Center of Radiology and Surgical Technologies, Ministry of Healthcare of Russian Federation, 197758 St. Petersburg, Russia; 6Ural Federal University, 620002 Yekaterinburg, Russia; 7Scientific Research Institute of Neurosciences and Medicine, 630117 Novosibirsk, Russia

**Keywords:** channelopathy, zebrafish, brain, ion channel, animal model

## Abstract

Channelopathies are a large group of systemic disorders whose pathogenesis is associated with dysfunctional ion channels. Aberrant transmembrane transport of K^+^, Na^+^, Ca^2+^ and Cl^−^ by these channels in the brain induces central nervous system (CNS) channelopathies, most commonly including epilepsy, but also migraine, as well as various movement and psychiatric disorders. Animal models are a useful tool for studying pathogenesis of a wide range of brain disorders, including channelopathies. Complementing multiple well-established rodent models, the zebrafish (*Danio rerio*) has become a popular translational model organism for neurobiology, psychopharmacology and toxicology research, and for probing mechanisms underlying CNS pathogenesis. Here, we discuss current prospects and challenges of developing genetic, pharmacological and other experimental models of major CNS channelopathies based on zebrafish.

## 1. Introduction

Membrane-bound ion channels play an important role in the development and functioning of multiple physiological systems, including the central nervous system (CNS) [1,2,3]. While transmembrane transport of K^+^, Na^+^, Ca^2+^ and Cl^−^ by these channels in the brain is key for the transmission of nerve impulses [1,4], experimentally induced (e.g., genetic or pharmacological) deficits of ion transport may lead to a wide range of systemic disorders termed ‘CNS channelopathies’ [2,5] (Table 1). The global prevalence of this disorder group is estimated as ~35 per 100,000 [3], with most common pathologies including epilepsy, followed by migraine, as well as various movement- and psychiatric disorders [5].

Animal models are a widely used tool for studying neurological diseases, including channelopathies, as well as their mechanisms and strategies for therapy [6,7]. Complementing rodent evidence, the zebrafish (*Danio rerio*) is rapidly gaining popularity as a translational model organism in biomedicine, including neurobiology and preclinical genetic and CNS drug screening [8,9]. Due to substantial genetic and physiological homology to humans, rapid growth and development, high fertility and medium-to-high throughput, zebrafish have become a powerful system for studying CNS pathologies [10,11]. The presence of all major mediator systems and neurotransmitters, as well as their receptors and transporters, in zebrafish makes them suitable for modeling of a wide range of CNS channelopathies. Zebrafish also demonstrate the ease of behavioral, genetic and pharmacological manipulations, hence further fostering the development of sensitive models of CNS channelopathies. Here, we discuss the prospects of using genetic, pharmacological and other experimental models of major CNS channelopathies based on zebrafish.

**Table 1 ijms-23-13979-t001:** Selected CNS channelopathies in humans.

Ion Channel and Associated Diseases	Symptoms	Gene	References
**K^+^**			
Episodic ataxia type I	Brief attacks of cerebellar ataxia, neuromyotonia	*KCNAI*	[4]
Andersen-Tawil syndrome	Ventricular arrhythmias, periodic paralysis, dysmorphic features, cognitive defects, memory impairments, depression, dysexecutive syndrome	*KCNJ2*	[12]
Nocturnal frontal lobe epilepsy type 5	Seizures during sleep, regular, sudden bicycle-pedaling leg and arm movements, shivering, feeling touchy, being pushed, falling and fear	*KCNT1*	[13]
Deafness, autosomal-dominant, type 2A	Symmetrical, delayed, progressive high-frequency hearing loss, tinnitus and vestibular dysfunctions	*KCNQ4*	[14]
Deafness, autosomal-recessive, type 4, with enlarged vestibular aqueduct	Inner ear malformation, high-frequency hearing loss, vestibular disorders	*KCNJ10*	[15,16]
Early infantile epileptic encephalopathy type 7	BFNC (benign clinical course), early seizures, hypotonia, dystonia, basal ganglia, thalamic hyperintensities	*KCNQ2*	[17]
Early infantile epileptic encephalopathy type 14	Partial seizures, microcephaly, developmental delay, intellectual disability, disability to eye contact and control head movement	*KCNT1*	[18,19]
EAST (Epilepsy, Ataxia, Sensorineural deafness, Tubulopathy)	Epilepsy, ataxia, sensorineural hearing loss, tubulopathy, intellectual disability, cerebellum changes, delayed motor and mental development, hypotonia	*KCNJ10*	[20]
Benign neonatal familial convulsions	Tonic posturing, clonic and generalized convulsions between days 1-14 of life	*KCNQ3*	[21]
Generalized epilepsy with paroxysmal dyskinesia	Absence seizure, mental retardation, generalized spike wave complexes, Lennox–Gastaut pattern in electroencephalogram (EEG), myoclonic seizures	*KCNMA1*	[22]
Spinocerebellar ataxia type 13	Delayed motor development, mental retardation, epilepsy, dysarthria, dysphagia, limb ataxia, impaired cognition	*KCNC3*	[23]
**Na^+^**			
Generalized epilepsy with febrile seizures plus (GEFS+)	Seizures with fever, including tonic, clonic, myoclonic or absence seizures, febrile seizures and febrile seizures plus, absences, myoclonic or atonic seizures, myoclonic-astatic epilepsy	*SCN1B, SCN1A*	[2,24]
Generalized epilepsy with febrile seizures plus type 2 (GEFS+2)	Generalized tonic/clonic convulsions, hyperexcitability	*SCN1A*	[25,26]
Benign familial infantile epilepsy	Partial seizures in occipito-parietal areas, slow deviation of head and eyes to one side, cyanosis, unilateral limb jerks	*SCN2A*	[27]
Cognitive impairment with or without cerebellar ataxia	Dementia, poor verbal memory and executive dysfunction, visuospatial dysfunction, intellectual impairment, limb ataxia, dysarthria, loss of proprioception, ataxia	*SCN8A*	[28]
Congenital indifference to pain, autosomal-recessive	Insensitivity to pain, absence of physiological responses to noxious stimuli	*SCN9A*	[29]
Small fiber neuropathy	Increased pain and temperature perception, pain (prickling, burning, shooting, and aching), cramps, restless legs, increased or decreased sweating, constipation or diarrhea, allodynia, hyperesthesia, pinprick sensation	*SCN9A*	[30]
Primary erythermalgia	Recurrent attacks of red, warm, and painful hands and/or feet, local red-purple discoloration, increased local temperature in hands/feet, burning pain	*SCN9A*	[31,32]
Potassium-aggravated myotonia	Muscle stiffness, cramps, muscle hypertrophy, respiratory dysfunction, painful myotonia, sensory disturbance, orbicularis oculi myotonia	*SCN4A*	[33,34]
Paroxysmal extreme pain disorder	Stiffening in flexion position, erythematous harlequin-type color changes, loss of consciousness, pain, tachycardia/bradycardia, bronchospasm, lacrimation; hypersalivation, rhinorrhea, syncopal events	*SCN9A*	[35]
Hyperkalemic periodic paralysis	Flaccid limb weakness, myopathy, muscle degeneration, myotonia, hyperkalemia	*SCN4A*	[36,37]
Hypokalemic periodic paralysis type 2	Episodes of flaccid paralysis, low serum K+, weakness of limbs, paroxysmal muscle stiffness	*SCN4A*	[38]
Early infantile epileptic encephalopathy type 11	Focal, tonic and tonic/clonic seizures, intellectual disability	*SCN2A*	[39]
Early infantile epileptic encephalopathy type 13	Pyramidal and extrapyramidal symptoms, developmental impairment, developmental and epileptic encephalopathy, multifocal epileptiform abnormalities, focal, generalized seizures, and epileptic spasms	*SCN8A*	[39]
Dravet syndrome	Prolonged and recurrent febrile seizures, hyperactivity, autistic and psychotic disorder, myoclonic and atypical absence, partial seizures	*SCN1A*	[40]
Paramyotonia congenita	Muscle weakness, myotonia in eyelids, neck and upper limb muscles	*SCN4A*	[41]
**Ca^2+^**			
Episodic ataxia type 2	Intermittent attacks of cerebellar, ataxia, migraine-like headache, vertigo, nausea, vomiting, dysarthria, tinnitus or EEG dysrhythmia, nystagmus		[42]
Hypokalemic periodic paralysis type 1	Quadriparesis, muscle weakness, hypokalemia	*CACNA1S*	[43,44]
Spinocerebellar ataxia type 6	Progressive cerebellar ataxia, gait ataxia, stiffness, cerebellar oculomotor deficit, upper limb ataxia, cerebellar dysarthria, dysphagia, tremor, mild bradykinesia with rigidity, mild peripheral neuropathy	*CACNA1A*	[45]
Episodic ataxia type 5	Vertigo, tinnitus, blurred vision, diplopia, myokymia, tremor, headache, unilateral numbness, weakness, loss of consciousness, migraine with aura, epilepsy, cerebellar ataxia	*CACNB4*	[46]
Familial hemiplegic migraine	Hemianopia, hemisensory loss, dysphasia, hemiparesis, weakness, migraine, attention and memory deficits	*CACNA1A*	[4]
Childhood absence epilepsy	Brief episodes of epileptic activity, depression, anxiety, typical absence seizures, impairment of consciousness without major motor symptoms, attention and cognitive impairments, dysgraphia	*CACNA1H*	[47,48]
Timothy syndrome	Long QT syndrome, autism, congenital heart defects, facial dysmorphisms, seizures, intellectual disability	*CACNA1C*	[49]
Åland Island eye disease (AIED, Forsius–Eriksson syndrome)	Fundus hypopigmentation, decreased visual acuity, nystagmus, astigmatism, color vision defect, progressive myopia, defective dark adaptation	*CACNA1F*	[50]
Cone-rod dystrophy, X-linked, type 3	Progressive dysfunction of photoreceptors (disturbed cone or cone-rod responses), diminished visual acuity, photophobia, myopia, central scotomas in visual fields, impaired color vision	*CACNA1F*	[51]
Congenital stationary night blindness type 2A	Myopia, hyperopia, nystagmus, strabismus, decreased visual acuity, impaired scotopic vision	*CACNA1F*	[52]
**Cl^−^**			
Thomsen’s disease	Mild myotonia, muscle hypertrophy, myalgia	*CLCN1*	[4]
Becker’s disease	Myotonia congenita, myotonia with the warm-up phenomenon, muscle hypertrophy, mild myopathy	*CLCN1*	[4]

## 2. A Brief Summary of CNS Channelopathies and Their Experimental Models

Channelopathies most commonly involve aberrant functions of calcium (Ca^2+^), potassium (K^+^), sodium (Na^+^) and chloride (Cl^−^) ion channels [1]. For example, K^+^ channels are one of the most abundant CNS channels, and include voltage-gated (Kv), inwardly rectifying (Kir), Ca^2+^-sensitive, ATP-sensitive and some other types [1]. Voltage-gated K^+^ channels are encoded by *KCNA (Shaker)*, *shal*, *KCNB (shab)*, *KCNC (shaw)*, *KCND*, *KCNE*, *KCNF*, *KCNG*, *KCNQ* and *KCNS* genes [1]. In humans, several K^+^ channel-related (e.g., *KCNA* and *KCNQ*) mutations are associated with CNS channelopathies, causing fatal brain deficits [53]. Mutations in *KNCQ2* are associated with severe epileptic encephalopathies [54,55], whereas some cases of the DEND (developmental delay, epilepsy and neonatal diabetes) syndrome are caused by mutations in *KCNJ11* [56], and benign familial neonatal convulsions (BFNC)—by the loss-of-function mutations in two K^+^ channel genes, *KCNQ2* and *KCNQ3* [57,58]. Likewise, the KCNA ion channel gene has a *Shaker* locus associated with CNS hyperexcitability in *Drosophila melanogaster* mutants, evoking spontaneous leg twitching due to aberrant CNS K^+^ fluxes [59,60,61]. Eight *Shaker* genes are also found in mammals (rats, mice and humans) [62,63]. The voltage-gated K^+^ channel Kv1.1, first discovered in mouse juxtaparanodal regions of myelinated axons and synaptic terminals [61,64], plays a role in repolarization, action potentials and the regulation of neurotransmitter release [1]. Kv1.1 knockout mice develop spontaneous epileptic activity due to the hyperexcitability of hippocampal cells [1]. Of the ten KCNA genes found in humans, four (*KCNA1*, *KCNA2*, *KCNA4* and *KCNA6*) are expressed in the brain, and *KCNA1* mutation causes episodic ataxia [1]. There are five genes of the KCNQ family (*KCNQ1*, *KCNQ2*, *KCNQ3*, *KCNQ4* and *KCNQ5*) encoding proteins forming different K^+^ channels. While the *KCNQ2* and *KCNQ3* gene products regulate the repetitive firing of neurons (when they are continuously exposed to a depolarizing stimulus), their mutations lead to BFNC [1].

Voltage-gated Ca^2+^ channels play an important role in the brain, regulating neuronal excitability and the release of neurotransmitters [65]. Ca^2+^ channels include L (long lasting), N (neuronal), P (Purkinje cell), Q (granule cell), R (toxin-resistant), and T (transient) types [1]. The L-, N-, P-, Q- and R- types of Ca^2+^ channels regulate neurotransmitter release [1], whereas mutations in the P- and Q-type channel genes cause ataxia and migraine in both human and animals, due to their high density in cerebellar Purkinje and granule cells that control locomotion [1]. Mutations in these channels commonly lead to overt epilepsy and ataxia phenotypes in several animal models. For example, mouse *leaner* mutants show seizures similar to human absence seizures, and progressive cerebellar degeneration caused by splicing mutation impact on the α subunit of the P/Q-type Ca^2+^ channel [1]. The *tottering* mutant mice also demonstrate absence-like seizures, whereas *rolling* mutants display no seizure activity, but poor limb coordination, falling and rolling over [1,66]. Mouse *lethargic* mutants exhibit absence-like seizures and ataxia without neurodegeneration, s*targazer* mice show spike-like wave seizures (common in absence seizures) in the cerebellum [1], and *Roker* mice present absence-like seizure and ataxia resembling the *tottering* mouse phenotype discussed above. Interestingly, ablating the P/Q-type Ca^2+^ channel in the serotonergic neurons increases aggression in male mice without affecting their anxiety, which not only implicates these channels in the regulation of some evolutionarily conservative behaviors (e.g., aggression), but suggests novel ways of alleviating aggressive behaviors in people with CNS channelopathies [67].

Deficient Na^+^ channels also play a key role in CNS pathogenesis [68]. The structure of the Na^+^ channels includes two subunits, the α-subunit encoded by SCNA genes (11 isoforms) and β-subunit encoded by SCNB genes (3–4 isoforms) [1]. Major CNS voltage-gated Na^+^ channel subtypes include Na_V_1.1, Na_V_1.2, Na_V_1.3, and Na_V_1.6 [69,70]. Na_V_1.1 channels are localized in the dendrites and cell bodies of excitatory neurons [71], Na_V_1.3 in neuronal soma [72], Na_V_1.2 in unmyelinated neuronal processes [72,73], and Na_V_1.6 in myelinated axons and dendrites [74]. The main CNS role of these channels is the regulation of the action potential in the soma, axons, and dendrites, thereby controlling overall neuronal excitability [75]. Mutated Na^+^ channels cause various types of epilepsy, including generic epilepsy with febrile seizures plus (GEFS+) [24] and GEFS+ type 2 (GEFS+2) [25], neuropathies [30], cognitive impairment [28], myotonia [33], and the Dravet’s syndrome [76] (see further).

Idiopathic or primary epilepsy is typically caused by mutations in genes encoding ion channels [1], including neuronal nicotinic acetylcholine receptor (nAChR) subunits, as well as voltage-gated K^+^ (Kv), Na^+^, and Ca^2+^ channels. A popular genetic rodent model of idiopathic epilepsy involves WAG/Rij rats (an outbred subline of *Wistar* rats) developed in the 1920s [77]. WAG/Rij rats can also be a valid genetic animal model of absence seizures [78], given their similar EEG and behavioral patterns with those in human absence seizures. Autosomal dominant nocturnal frontal lobe epilepsy (ADNFLE), BFNC and severe myoclonic epilepsy in infancy (SMEI) are common types of idiopathic epilepsy [79], whose candidate genes include *CACNA1A*, *CACNA1G*, *CACNA1H*, *CACNA1I*, *CACNAB4*, *CACNAG2* and *CACNG3* that encode subunits of Ca^2+^ channels, and *GABRG2*, *GABRAB3*, *GABRA5*, *GABA(B1)* and *GABA(B2)* that encode subunits of the Cl^−^ channels on the gamma aminobutyric acid (GABA)-A receptors [80], respectively.

Zebrafish have become popular model organisms in translational neuroscience, physiology and preclinical screening mostly because of their low cost, easy maintenance, as well as high genetic and physiological homology with mammals, including humans [81]. Fast development and transparent embryos make zebrafish a useful tool for genetic manipulations, with multiple established genetic models of CNS disorders, including autism [82], schizophrenia [83], attention deficit hyperactivity disorder (ADHD) [84] and epilepsy [81]. Zebrafish are also susceptible to a wide range of pharmacological agents, including a conventional GABA-lytic drug pentylenetetrazole (PTZ) that blocks Cl^−^ ionophore on GABA-A receptors and induces seizures in zebrafish that resemble those observed in mammals [85].

One of the best-characterized genetic epilepsy models in zebrafish is presently an N-ethyl-N-nitrosourea (ENU)-generated model of Dravet’s syndrome, involving mutation in the SCN1A gene that affects voltage-gated Na^+^ channels [86] (Table 2). For example, the *scn1Lab* zebrafish mutants have spontaneous seizures that resemble those in patients with Dravet’s syndrome, both behaviorally and electrophysiologically [85]. Another interesting zebrafish model of epilepsy is based on the *kcnj10* morphants, recapitulating the EAST (Epilepsy, Ataxia, Sensorineural deafness, Tubulopathy) syndrome [87]. *KCNL10* encodes a K^+^ channel protein that regulates K^+^ levels in glial cells and hence modulates their excitability [88]. Other promising candidates for genetic models of epilepsy in zebrafish involve temperature-gated ‘transient receptor potential vanilloid’ (TRPV) TRPV1 and TRPV4 ion channels. TRPVs are a family of ion channels with high Ca^2+^ permeability, thus regulating neuronal excitability in both mammals [89,90] and zebrafish [91]. Activation of thermosensitive TRPV1 and TRPV4 ion channels generally increases pathological excitability of neurons and causes febrile seizures in zebrafish [92].

Ataxia is another complex neurological symptom that involves poor coordination of movements or gait, and some forms of which are commonly caused by channelopathies. For example, ataxia is observed in both clinical EAST syndrome and in zebrafish knockdown of *kcnj10*, which presents as abnormal swimming, frequent spontaneous tail flicks, circling, bursts of speed and other aberrant movements [102]. Spinocerebellar ataxia type 6 (SCA6) is a neurological disease resulting from a mutation in *CACNA1A* [103]. Due to teleost-specific duplication of the zebrafish genome, there are two gene copies encoding the CaV2.1 Ca^2+^ channels, *cacna1aa* and *cacna1ab*. Two mutations of the *cacna1ab* gene include *tb204a* [104] and *fakir* [105], both disrupting Ca^2+^ channel function in zebrafish [106]. The *tb204a* mutation (which replaces tyrosine with asparagine (Y1662N) at the carboxyl end of the CaV2.1a channel) reduces the mobility of zebrafish larvae and lowers intracellular Ca^2+^ in presynaptic neuromuscular junctions [104]. The *fakir cacna1ab* mutation (referred to as L356V, substituting leucine 356 with a valine), impairs larval locomotion and causes depolarizing shifts during the Ca^2+^ channel activation [105]. Although both *cacna1a* and *-b* mutations in zebrafish have similar effects on locomotion, there are some differences in the mechanism of occurrence of these deficits: while *fakir* mutants have decreased touch sensitivity and altered transmission between motor neurons and slow-twitch fibers in muscles [107], *tb204a* mutant fish demonstrate aberrant transmission in neuromuscular junction [104].

Spinocerebellar ataxia type 13 (SCA13) is neurodegenerative disease caused by mutated *KCNC3 gene encoding the* Kv3.3 K^+^ ion channel, presenting as the atrophy of cerebellar neurons, motor dysfunction and intellectual disability followed by progressive ataxia in adulthood [108,109,110]. In zebrafish, *kncn3* mutant larvae also display reduced startle response to touch [108], supporting the validity of this model of ataxia. As already noted, the Dravet’s syndrome is a severe childhood epileptic encephalopathy due to a mutation in the *SCN1A* gene encoding voltage-gated Na^+^ channels (Nav1.1) [111]. In addition, the *SCN2A, SCN8A, GABRA1* and *STXBP1* genes (that encode Na^+^, Cl^−^ channels and syntaxin-associated proteins, respectively) also play a role in Dravet’s pathogenesis [76,112]. Mutations within the *SCN1A* gene (encoding the Nav1.1 channel) disrupt Na^+^ flow into neurons [113], hence affecting the action potential of the GABA-ergic neurons in brain structures that are critical for modulating seizure activity (e.g., in cortex and the hippocampus) [114]. The *scn1lab* zebrafish mutants show good survival rate, increased synaptic activity and reduced inhibitory tone in the brain [115]. In addition, by 7th day post-fertilization (dpf), the *scn1lab^mut/mut^* fish show fewer GABA-ergic neurons and increased glutamatergic signaling [115]. Interestingly, the number of glial cells in these mutants is also markedly higher, which may be useful for studying (and, eventually, parsing) the role of glia in the pathogenesis of Na^+^ channelopathies [115].

## 3. Towards Valid Zebrafish Models Relevant to CNS Channelopathies

Overall, ion channels play a critical role in functioning and control of various physiological systems, including CNS [1] (Table 1). Genetic mutations of K^+^, Na^+^, Ca^2+^ and Cl^−^ channels in neurons lead to severe cardiovascular, respiratory and CNS diseases [1,4] (Table 1, Table 2, Table 3 and Table 4). The search for novel treatments for these prevalent and debilitating maladies represents an urgent unmet biological problem, necessitating reliable, throughput, sensitive and efficient animal models. As already discussed, the use of zebrafish to model CNS channelopathies is beginning to develop rapidly in translational research, also offering an indispensable and effective tool for rapid CNS drug screening. Indeed, zebrafish are highly sensitive to a wide range of major CNS drugs, including antidepressant [116], antiepileptic [117], antipsychotic [118], anxiolytic [119], nootropic [120] and other psychoactive substances [121,122], including addictive drugs, such as alcohol [123,124], nicotine [123,125], caffeine [126] and synthetic salts [127,128,129]. The low cost of maintenance, easy breeding, fast development cycle, the availability of “two models in one” (i.e., both larvae and adult fish), as well as the simplicity of treatment with acute and chronic substances, all make this model organism an indispensable tool for drug discovery and high-throughput preclinical screening [130].

A unique practical advantage of using zebrafish for pharmacological modeling of epilepsy-related phenotypes of channelopathies is the possibility of treating multiple fish with a water-diluted convulsant drug at once, evoking simultaneous and rapid development of seizures in these zebrafish and allowing a rapid screening of a wide range of anticonvulsants at the same time [148]. For example, as we calculated based on our own 20-year experience with experimental epilepsy models in both mammals and fish, a typical time-consuming full-day rodent experiment with systemic (e.g., intraperitoneal, i.p.) PTZ injection to induce seizures would require ~10 times more time than a zebrafish study of a similar experimental design and sample size, where the drug is administered via water immersion to multiple fish at once, and several fish are recorded per trial, to assess their seizure-specific locomotor endpoints. The simplicity of pharmacological model of epilepsy also highlights the similarity of the observed phenotypes of zebrafish seizure behavior (i.e., similar clonic and tonic–clonic seizures, hyperlocomotion, corkscrew-like circular swimming and ataxia) to those in rodent models [149]—as discussed above, another highly relevant phenotype to modeling CNS channelopathies.

Furthermore, with the additional genome duplication event in teleost fishes, the zebrafish becomes an excellent model for various genetic manipulations and modifications, including the creation of morphants and knockout mutants of known human CNS diseases associated with specific mutations [81,150,151], including those affecting ion channels in the brain [152,153]. An important conceptual advantage of zebrafish models in this regard is that with partially duplicated genome, it becomes possible to create mutants whose genetic analogues in rodents are impossible due to the lethality of these mutations [130] (a situation not uncommon for mutations in genes encoding critical ion channels with so many key physiological functions). This aspect paves the way for the creation of more accurate model systems for studying the pathogenesis of the development of mutations in ion channels in aquatic vertebrates with 70+% genetic similarity to humans [154].

Furthermore, behavioral phenotypes of zebrafish are quite simple, clear-cut and generally comparable to rodents in all major domains [155]. There are also multiple behavioral tests and paradigms to assess zebrafish motor phenotypes, similarly to rodents, hence making behavioral phenotyping in general (including targeting epilepsy and, likely, other related CNS channelopathies) a simple, fast and efficient procedure [149,155]. In addition, zebrafish models seem to be more resilient (than rodent models) to variation in external testing procedures, which may contribute to increased reproducibility and reliability of laboratory data using zebrafish [156].

Interestingly, the use of a 3D system software for studying zebrafish behavior [157] enables the creation of libraries of locomotor responses to various pharmacological agents [155]. Despite the simplicity of demonstrated phenotypes, zebrafish have complex behavioral responses to various stimuli, including social behavior in a group, sufficient to study and model complex behavioral paradigms [158]. Developing high-efficiency software, along with 3D behavioral tracking, facilitates automated behavioral screening of chemical agents using artificial intelligence (AI) [159]. It is therefore likely that AI-based drug screening in zebrafish, including tracking of their tonic and clonic seizures in epilepsy simulation, will greatly advance preclinical studies of anticonvulsants and other drugs to treat channelopathies. Zebrafish also present several protocols to assess CNS channel activity or action potentials using various electrophysiological methods [160,161,162]. For instance, electrophysiological characterization of heterologously expressed zebrafish Kcnh1a and Kcnh1b channels reveals functional patterns similar to those of human K^+^ channels [163]. Such methods, particularly useful and feasible in zebrafish with their smaller transparent brains, enable the evaluation of the action potential and direct linking aberrant (e.g., mutant) ion channel activity to CNS disorders.

Despite multiple obvious advantages, zebrafish models also have significant limitations as well. For example, no animal model is an exact reproduction of complex human CNS diseases [164]. While there is a large number of genetic models of CNS diseases in zebrafish, these animals are often unable to reproduce due to the loss of fertility. Another limitation of zebrafish models is the methods of administering the test substances, mainly by water immersion or i.p. injection, which complicates studying water-insoluble compounds and requires the use of special solvents [165]. Another important technical limitation of zebrafish as a model is the likely difference in dosing between humans and fish, which requires a specific recalculation of the used dose of the drug for preclinical studies when translating the results to humans [166,167].

The reproduction of core clinical symptoms in rodents and zebrafish represents yet another key translational issue with modeling CNS channelopathies in fish. For example, one of broad clusters of clinical signs of CNS channelopathies (beyond epilepsy) is migraine and headaches of various types and etiologies (Table 1) [4]. Indeed, K^+^ channels are found mainly in the heart and brain, where they modulate and stabilize action potentials of cells by regulating K^+^ current [168]. The loss of ion channel function due to mutation of the *KCNK18* gene is a common cause of migraine [169]. In addition to K^+^ channels, mutation in *CACNA1A* Ca^2+^ channel gene contribute to the pathogenesis of type 2 episodic ataxia and hemiplegic migraine [170]. First, assessing the level of headache and the ability to accurately diagnose this condition in animal models are a challenging task [171]. For instance, in genetic models of migraine-like conditions in rodents, it is impossible to accurately assess the presence and the severity of headache. Thus, altered sensitivity of animals to various stimuli is rather tested instead, similar to clinical signs of migraine, such as photophobia, increased tactile and sound sensitivity, and preference for darkness.

It is therefore impossible to assume that migraine-like pain models in rodents cause migraines per se, and the same problem applies a priori to any such zebrafish models as well. In addition, when modeling migraine, only that accompanied by the aura lends itself to quantitative and qualitative assessment. In contrast, it seems that migraine without the aura cannot be reliably diagnosed in animal models. Accordingly, it remains unclear whether it is possible to simulate migraine and migraine-like pain in fish, and whether fish are even capable of experiencing headaches similar to those observed clinically. Likewise, we do not know what pharmacological and genetic tools can help model these conditions best in both rodents and fish.

In summary, the high global prevalence of diseases directly related to abnormal ion channels necessitates our improved understanding of the pathological mechanisms underlying CNS channelopathies, as well as the development of novel pharmacological agents for their treatment. For this, zebrafish models can be a useful tool for identifying evolutionary-conservative mechanisms underlying CNS channelopathies, and a powerful efficient drug-screening platform, to induce or prevent these conditions in vivo. With multiple conceptual and methodological questions remaining open in the field (Table 5), this collectively calls for a wider use of zebrafish models in studying CNS channelopathies. 

## Figures and Tables

**Table 2 ijms-23-13979-t002:** Summary of selected genetic causes of CNS channelopathies.

Pathology	Na^+^	K^+^	Ca^2+^	GABA_A_	Nicotinic	Cl^−^	Glycine	References
Epilepsy	SCN1A	KCNQ2	CACNA1H	GABRA1	CHRNA2			[2,17,21,24,25,26,47,48,80,93]
	SCN1B	KCNQ3		GABRB3	CHNRA4			
	SCN2A	KCNMA1		GABRG2	CHRNB2			
Migraine	SCN1A		CACNAA1A					[94,95]
Myasthenia Fetal akinesia					CHRNA1			[96]
					CHRNB1			
					CHRNG			
					CHRND			
					CHRNE			
Myotonia	SCN4A					CLCN1		[4,33,34]
Periodic paralysis	SCN4A	KCNJ2	CACNA1S					[12,38,97]
Pain, Erythema	SCN9A							[35]
Ataxia		KCNA1	CACNA1A					[4,23,45]
		KCNC3						
Hypereplexia							GLRA1	[98,99]
							GLRB	
Dravet syndrome (DS)	SCN1 SCN8							[100,101]
Spino-cerebellar ataxia type 13 (SCA13)		KCNC3						[23]

**Table 3 ijms-23-13979-t003:** Summary of rodents’ models of CNS channelopathies.

Animal Model	Ion Channel Subtype	Phenotype	References
** *Epilepsy* **			
Tottering mice	Ca^2+^	Absence-like seizures, abnormal eye movements, dystonia	[1,66,131,132,133]
Leaner mice	Ca^2+^	Absence ataxia and progressive cerebellar degeneration	[1]
Rolling Nagoya mice	Ca^2+^	No seizures, limb incoordination, falling and rolling over	[1]
Lethargic mice	Ca^2+^	Absence epilepsy and ataxia without neurodegeneration	[1]
Stargazer mice	Ca^2+^	Spike-like wave seizures characteristic of absence epilepsy parallel with obstruction in the cerebellum and inner ear	[1]
Roker mice	Ca^2+^	Intermittent seizures similar to absence epilepsy, mild ataxia, reduction in branching of the dendritic arbor	[65,134]
WAG/Rij rats	Ca^2+^, K^+^, Na^+^	Facial myoclonic jerks, eye and vibrissae twitching, head tilting	[135,136]
*Ducky* mice	Ca^2+^	Cerebellar, medullar and spinal atrophy, spike-wave phenotype, abnormal morphology of Purkinje neurons, cerebellar dysfunction	[137,138]
* **Timothy syndrome** *			
TS2-like mouse	Ca^2+^	Markedly restricted, repetitive, and perseverative behavior, altered social behavior, altered ultrasonic vocalization, enhanced tone-cued and contextual memory following fear conditioning	[139]
** *Ataxia* **			
Pingu (Pgu) mice	K+	Abnormal gait, smaller body size, splayed hind limbs, growth delayed, motor incoordination, flattened body posture, severe tremors, myoclonic jerks, ataxic movement	[140]
Tottering mice	Ca^2+^	Episodic attacks of ataxia due to stress	[141]
Leaner mice	Ca^2+^	Degeneration of differentiated granule, Golgi, and Purkinje cells in cerebellum	[141]
Rolling Nagoya mice	Ca^2+^	Severe ataxia without motor seizures	[141]
V408A/+ mice	K^+^	Loss of motor coordination, increased frequency and amplitude of spontaneous GABAergic inhibitory postsynaptic currents (IPSCs), motor impairment,	[142]
*Ducky* mice	Ca^2+^	Ataxia, and paroxysmal dyskinesia, “ducky” gait	[143]
* **Migraine** *			
*S218L mutant mice*	Ca^2+^	Severe migraine with seizures, coma, severe cerebral edema following mild head trauma, propensity to multiple cortical spreading depression (CSD) events	[68]
*R192Q knock-in mice*	Ca^2+^	Increased glutamate release in the cerebellar cortex, decreased threshold for CSD, decreased neuronal response to nociceptive activation, photophobia; unilateral head grooming; lateralized winking/blinking, migraine-like headache	[144,145,146]
*S218L knock-in mice*	Ca^2+^	Exquisite sensitivity to CSD (reduced triggering threshold, increased propagation velocity and frequency of CSC events), ataxia, seizures, migraine-like headache, photophobia; unilateral head grooming; lateralized winking/blinking	[145,146]
*Atp1a2^+/R887^* mice	Na,K-ATPase	Decreased induction threshold, increased velocity of propagation CSD, higher susceptibility to fear and anxiety	[147]

**Table 4 ijms-23-13979-t004:** Selected zebrafish models of CNS channelopathies.

CNS Disorder	Modulated Ion Channel	Zebrafish Model	Phenotypes	References
Dravet’s syndrome	Na^+^	*scn1Lab* mutants	Seizures	[85]
		*scn1lab* mutants	Increased synaptic activity and reduced inhibitory tone in the brain	[115]
Epilepsy	Ca^2+^	*kcnj10* morphants	Epilepsy, ataxia and sensorineural deafness	[87]
	Ca^2+^	*kcnj10* knockdown	Abnormal swimming, frequent spontaneous tail flicks, circling, bursts of speed and other aberrant movements	[102]
Spinocerebellar ataxia type 6	Ca^2+^	*tb204a* mutants	Reduced the locomotor activity, accompanied by reduced intracellular Ca^2+^ in presynaptic neuromuscular junctions	[104]
Spinocerebellar ataxia type 13	K^+^	*kcnc3* mutants	Reduced startle response to touch	[108]

**Table 5 ijms-23-13979-t005:** Selected open questions related to zebrafish-based modeling of CNS channelopathies.

* Open Questions *
Given generally high overall genetic homology (70%) of zebrafish to humans, do CNS channelopathies share similarly overlapping genetic substrate between the two model organisms? Zebrafish have a generally high (85+%) percentage of gene orthologs for known human-disorder-associated genes. Is there a similar (i.e., comparably high) percentage of disorder-associated genes for CNS channelopathies? What is the percentage of unique, human-specific channelopathies that are not present in zebrafish, and vice versa? What is the exact impact of teleost fish-specific genome duplication on the expression of disordered phenotypes in CNS channelopathies? What are reliable behavioral or neurological tests that can distinguish between mental and neurological manifestations of CNS channelopathies in experimental animal models in general, and in zebrafish in particular? Are there overt sex and strain differences in the pathogenesis of CNS channelopathies in zebrafish models? Is there a direct correlation between physiological aspects of CNS channelopathies (e.g., Ca^2+^ channel modulation) and behavioral phenotypes (e.g., seizure)? If so, is this correlation translational (e.g., from zebrafish to humans)? How can specific diet (e.g., high-Na^+^ or low-Ca^2+^ diet) influence various phenotypes of CNS channelopathies? Can other dietary factors (e.g., high-fat/high-sugar diet) play a role in CNS channelopathies? What non-pharmacological therapies can be effective for CNS channelopathy treatment? Are there epigenetic mechanisms underlying CNS channelopathies in zebrafish models? Can certain animal models (e.g., based on chronic pain) undergo epigenetic modulation to trigger the development of CNS channelopathies in their progenies? Does early-life stress or early-life pain exposure influence CNS channelopathies? Is it possible to assess migraine-like pain in zebrafish? What assessment methods (and putative specific behavioral indices) may help diagnose migraine-like condition in zebrafish?

## Data Availability

Not applicable.

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
