# Peer review of "Towards Zebrafish Models of CNS Channelopathies"

_ijms, 2022, doi:10.3390/ijms232213979_

Round 1

Reviewer 1 Report

In this review article, authors discussed unique advantages and limitations of Zebrafish as an experimental animal model for CNS disorders.

Major comment:

Missing Zebrafish electrophysiology (what action potential looks like, similarity compared to human brain APs, roles of Na+, K+, HCN1 in these APs) that link ion channel mutations and CNS disorders.

Minor comments:

1)    Page 2 (in table 1): SLC26A4 and SCN8A are NOT K channels,

2)    Page 4, under “General overview of CNS channelopathies”, HCN genes do not encode voltage-gated K+ channels. HCN channels are permeable to both K+ and Na+ ions. At the hyperpolarized membrane potential when the channel is activated, more Na+ than K+ ions pass through the HCN channels due to much larger driving force for Na+ and for K+.

3)    Outdated Reference 1: Li, M. and H.A. Lester, Ion channel diseases of the central nervous system. CNS drug reviews, 2001. 7(2): p. 214-240.

For example, right after HCN genes, authors stated “In humans, only two K+ channel-related (KCNA and KCNQ) mutations are associated with CNS channelopathies, causing fatal brain deficits [1].” But in table 2 (CNS channelopathies), authors included HCN1 under K+, although it should not be grouped under K+ channels.

A 2016 review of neurological channelopathies has included more K+ channel-related mutations, Genetic neurological channelopathies: molecular genetics and clinical phenotypes | Journal of Neurology, Neurosurgery & Psychiatry (bmj.com).

4)    Page 4: “The KCNA ion channel has a Shaker locus associated with CNS hyperexcitability “in in Drosophila” mutants, evoking spontaneous leg twitching due to aberrant CNS K+ fluxes [12-14].” – grammar issue. Please check grammar issues using word features.

5)    Superscript issue: there are many places superscript issue exists. For example, Page 5: “Voltage-gated Ca2+ channels play an important role in the brain, regulating neuronal excitability, neurotransmitter release, and “Ca2+ “ influx into neurons [18].” Also, in table 3…in page 11…

6)    Table 2: HCN1 should not be listed under K+ channel group.

Author Response

Dear Reviewer 1, thank you very much for your evaluating of our manuscript ijms-1943705, by Kolesnikova et al. As requested, please find attached our revised R1 manuscript and the point-by-point rebuttal letter that addresses the Reviewer’ concerns. For your convenience, all revised items are now highlighted in yellow in the resubmitted MS text file.

Point 1. In this review article, authors discussed unique advantages and limitations of Zebrafish as an experimental animal model for CNS disorders. Missing Zebrafish electrophysiology (what action potential looks like, similarity compared to human brain APs, roles of Na+, K+, HCN1 in these APs) that link ion channel mutations and CNS disorders.

Reply: The authors thank the expert Reviewer very much for their generally positive evaluation of our MS. As recommended, we have now modified the MS accordingly, to insert data on zebrafish electrophysiology that link ion channel mutations and CNS disorders.

Point 2. Page 2 (in table 1): SLC26A4 and SCN8A are NOT K channels,

Reply: Corrected, as requested.

Point 3.  Page 4, under “General overview of CNS channelopathies”, HCN genes do not encode voltage-gated K+ channels. HCN channels are permeable to both K+ and Na+ ions. At the hyperpolarized membrane potential when the channel is activated, more Na+ than K+ ions pass through the HCN channels due to much larger driving force for Na+ and for K+.

Reply: As recommended, we have now modified the MS to alter the sentence, aiming at a greater clarity.

Point 4.   Outdated Reference 1: Li, M. and H.A. Lester, Ion channel diseases of the central nervous system. CNS drug reviews, 2001. 7(2): p. 214-240.

Reply: As recommended, we have updated the reference.

Point 5. Right after HCN genes, authors stated “In humans, only two K+ channel-related (KCNA and KCNQ) mutations are associated with CNS channelopathies, causing fatal brain deficits [1].” But in table 2 (CNS channelopathies), authors included HCN1 under K+, although it should not be grouped under K+ channels.

Reply: Corrected, as requested.

Point 6. A 2016 review of neurological channelopathies has included more K+ channel-related mutations, Genetic neurological channelopathies: molecular genetics and clinical phenotypes | Journal of Neurology, Neurosurgery & Psychiatry (bmj.com).

Reply: As recommended, we have now modified the MS accordingly, to include more K+ channel-related mutations from suggested reference by the expert Reviewer.

Point 7. Page 4: “The KCNA ion channel has a Shaker locus associated with CNS hyperexcitability “in in Drosophila” mutants, evoking spontaneous leg twitching due to aberrant CNS K+ fluxes [12-14].” – grammar issue. Please check grammar issues using word features.

Reply: Corrected, as requested.

Point 8. Superscript issue: there are many places superscript issue exists. For example, Page 5: “Voltage-gated Ca2+ channels play an important role in the brain, regulating neuronal excitability, neurotransmitter release, and “Ca2+ “ influx into neurons [18].” Also, in table 3…in page 11…

Reply: Corrected, as requested.

Point 9. Table 2: HCN1 should not be listed under K+ channel group.

Reply: Corrected, as requested. Overall, the authors believe that in the revised MS we have now carefully addressed all concerns and comments made by the expert Reviewer. We shall be grateful if you could consider our revised MS for publication in your Journal.

Reviewer 2 Report

Dear Authors,

your manuscript about zebrafish models of CNS Channelopathies is complete and exhaustive. IT is well wirtten and I have no comments to do about its scientific soundness. All possible pathologies and gene involved are well described and can be mimicked in zebrafish model. I really appreciate also the description of animal model differences and the reasons why zebrafish could be more useful then mammals.

Regards

Author Response

Dear Reviewer 2, thank you very much for your evaluating of our manuscript ijms-1943705, by Kolesnikova et al. As requested, please find attached our revised R1 manuscript and the point-by-point rebuttal letter that addresses the Reviewer’ concerns. For your convenience, all revised items are now highlighted in yellow in the resubmitted MS text file.

Point 1. The manuscript about zebrafish models of CNS Channelopathies is complete and exhaustive. IT is well written and I have no comments to do about its scientific soundness. All possible pathologies and gene involved are well described and can be mimicked in zebrafish model. I really appreciate also the description of animal model differences and the reasons why zebrafish could be more useful than mammals.

Reply: The authors thank the expert Reviewer very much for their positive evaluation of our MS.

Reviewer 3 Report

Table 2 - The addition of references on which the information is based is necessary

The title “Discussion” is not appropriate for subchapter, because the manuscript is not comprehensive review with typical subchapters. I suggest changing the title to reflect the information contained in this part of the article

In summary, the importance of zebrafish in models of CNS channelopathies should be more clearly emphasized

The authors should consider adding some graphs or diagrams (e.g. regarding the using of zebrafish in models of channelopaties) to the article. This would make the article more attractive to the reader

Table 1 and chapter “General overview of CNS channelopathies” do not correspond directly to the title. I suggest shortening this part of the article

Author Response

Dear Reviewer 3, thank you very much for your evaluating of our manuscript ijms-1943705, by Kolesnikova et al. As requested, please find attached our revised R1 manuscript and the point-by-point rebuttal letter that addresses the Reviewer’ concerns. For your convenience, all revised items are now highlighted in yellow in the resubmitted MS text file.

Point 1. Table 2 - The addition of references on which the information is based is necessary

Reply: As recommended, we have modified Table 2, to insert the references on which the information is based.

Point 2. The title “Discussion” is not appropriate for subchapter, because the manuscript is not comprehensive review with typical subchapters. I suggest changing the title to reflect the information contained in this part of the article

Reply: As recommended, we have now replaced “Discussion” with “Zebrafish to model CNS channelopathies”, to attend to the expert Reviewer’s comments.

Point 3. In summary, the importance of zebrafish in models of CNS channelopathies should be more clearly emphasized.

Reply: As recommended, we have modified the MS to bettr emphasize the importance of zebrafish as models of CNS channelopathies.

Point 4. The authors should consider adding some graphs or diagrams (e.g., regarding the using of zebrafish in models of channelopathies) to the article. This would make the article more attractive to the reader

Reply: The authors thank Reviewer 3 for this suggestion. In principle, we are always all for adding some visual graphic illustrations to the scientific texts. In fact, almost all our papers have some graphs. However, in this particular case, we opted not to insert graphs or diagrams in MS because we believe that we would be reintroducing the information already presented in MS and, thus, may overload the reading. We hope the expert Reviewer would agree with such reasoning in this particular case.

Point 5. Table 1 and chapter “General overview of CNS channelopathies” do not correspond directly to the title. I suggest shortening this part of the article

Reply: As recommended, we have modified Table 1 and the subheadings accordingly, attending to the review concerns. Overall, the authors believe that in the revised MS we have now carefully addressed all concerns and comments made by the expert Reviewer. We shall be grateful if you could consider our revised MS for publication in your Journal.

Sincerely,

Murilo S. de Abreu, Ph.D.

Allan V. Kalueff, Ph.D.